# Disability and Access to Sexual and Reproductive Health Services in Cameroon: A Mediation Analysis of the Role of Socioeconomic Factors

**DOI:** 10.3390/ijerph16030417

**Published:** 2019-02-01

**Authors:** Pierre DeBeaudrap, Charles Mouté, Estelle Pasquier, Muriel Mac-Seing, Pulchérie U. Mukangwije, Gervais Beninguisse

**Affiliations:** 1Centre Etude en Population (CEPED), Institut de Recherche pour le Développement, Université Paris Descartes, INSERM 1244, 75006 Paris, France; charlesmoute@gmail.com; 2Institut de Formation et de Recherche Démographiques (IFORD), Yaounde BP1556, Cameroon; gbeninguisse@yahoo.fr; 3Expertise France—5% Initiative for HIV, Malaria and TB, 75006 Paris, France; estellepasquier9@gmail.com; 4School of Public Health, University of Montreal, Montreal, QC H3N 1X9, Canada; muriel.k.f.mac-seing@umontreal.ca; 5Humanity & Inclusion, 69371 Lyon, France; p.mukangwije@hi.org

**Keywords:** disability, sexual and reproductive health, access to health services, Sub-Saharan Africa, mediation analysis, epidemiology

## Abstract

There is growing evidence showing that people with disabilities face more frequently socioeconomic inequities than their non-disabled peers. This study aims to examine to what extent socioeconomic consequences of disability contribute to poorer access to sexual and reproductive health (SRH) services for Cameroonian with disabilities and how these outcomes vary with disabilities characteristics and gender. It uses data from a population-based survey conducted in 2015 in Yaounde, Cameroon. Mediation analysis was performed to determine how much of the total association between disability and the use, satisfaction and difficulties to access SRH services was mediated by education level, material wellbeing lifetime work participation and availability of social support. Overall, disability was associated with deprivation for all socioeconomic factors assessed though significant variation with the nature and severity of the functional limitations was observed. Lower education level and restricted lifetime work mediated a large part of the association between disability and lower use of HIV testing and of family planning. By contrast, while people with disabilities reported more difficulties to use a SRH service, no mediating was identified. In conclusion, Cameroonians with disabilities since childhood have restricted access to SRH services resulting from socioeconomic factors occurring early during the life-course.

## 1. Introduction

An estimated one billion of people in the world have some sort of disabilities and 80% of them live in limited-resource settings [1]. There is now growing evidence showing that people with disabilities experience poorer health outcomes than those without disabilities and are still left behind [1,2,3]. There is also extensive evidence from both high and low-resource settings that people with disabilities face more frequently socioeconomic inequities than their non-disabled peers, which include higher rate of poverty, lower education level and increased rate of unemployment [1,4,5,6,7,8,9,10]. As these social disparities are in turn strong determinants of health inequities, including restricted access to health services and poor health outcomes [11,12,13], part of the association between disability and adverse health outcomes may be mediated by the socioeconomic disadvantages [14,15].

Sexual and reproductive health (SRH) is an essential component of health and a pillar for sustainable development [16] that deserve important focus in many resource-limited countries as unsafe sex, abortions, HIV (human immunodeficiency virus) and other sexual transmitted infections, sexual and gender-based violence and other pregnancy-related adverse outcomes constitute major risk factors of mortality and morbidity in these settings [17,18]. Yet, the needs of people with disabilities in this area have been neglected for decades in the first place because of the widespread view that they are not sexually active [19]. Consequently, there is a lack of evidence on their SRH outcomes that could be used to inform policy makers, donors, actors and beneficiaries, and to guide interventions. Most studies involving people with disabilities have looked at access to antenatal care or pregnancy outcomes and were conducted in high-resource settings [1,20]. In resource-limited context, a large cross-sectional survey conducted in Sierra Leone found that, after adjusting for socioeconomic factors, women with disabilities did not report more difficulties to access maternal services than their non-disabled peers [21]. In a qualitative study conducted in Nepal difficulties to access maternal care seemed to be more related to poverty than disability [22]. By contrast, qualitative works conducted in Zambia, Uganda and Cameroon found that women with severe disabilities faced also attitudinal barriers to access to maternal health care [23,24,25]. A recent large-scale quantitative survey conducted in four countries from Southern Africa reported that the main barriers to health care were lack of transport, availability of services, inadequate drugs or equipment and costs and found that both socioeconomic status and disability were associated with the likelihood of experiencing these barriers [26]. In both low and high-resource settings, difficulties in the interactions with health workers have been frequently reported [1,23,27,28,29].

The differences between studies in the levels and nature of difficulties to access SRH care reported by people with disabilities may be due to the different methodological approaches used and to differences in the characteristics of the population investigated or in the study contexts. While this confirms that more data is needed to better understand the situation of people with disabilities and their difficulties to access SRH services, it also shows that a careful consideration should be given to the importance of socioeconomic factors as mediating factors and to the heterogeneity of the population with disabilities [30].

The present study aims to investigate the relationships between disability and access to and use of SRH services among people with disabilities since childhood, using data from a survey conducted in Cameroon in 2015. More specifically, it examines to what extent socioeconomic consequences of disability (lower education level, deprivation, absence of work, restricted social network) contribute to poorer access to sexual and reproductive health services and how outcomes vary with people with disabilities’ type of limitation, sex and age.

## 2. Materials and Methods

### 2.1. Study Design and Participants

This population-based cross-sectional study took place in Yaoundé between 2 October 2014 and 30 November 2015. A multistage sampling strategy was used to randomly select people with disabilities and matched controls from households of the general population (Appendix A). All people aged 15 to 49 years with severe difficulties in at least one domain or with some difficulties in at least two domains of the Washington Group Short Set (WGSS) questionnaire for ≥12 months were considered as living with disabilities and eligible for the study [31]. This tool includes a small number of questions covering six functional domains or basic actions: seeing, hearing, walking, cognition, self-care and communication. Each question asks the respondent to rate on a four-point scale how much difficulty he/she has experienced in the domain (Appendix B). For each person with a disability included in the study, a control of similar age and sex, living in the same enumeration area but in a different household and not meeting the functional limitation criteria was recruited. For this analysis, the study population was restricted to participants with physical and/or sensory (visual and hearing) difficulties that occurred before the age of 10 years (and their matched controls) in order to ensure a chronological sequencing of disability and of the potential mediating factors assessed.

### 2.2. Procedures

Face-to-face structured interviews were conducted at the home of the eligible subjects to collect data on their life-course history of employment, resources, sexual partnership and fertility using the life-grid method [32], on their activity limitations, on their knowledge on HIV and family planning and on their use of family planning, HIV testing and main other SRH services (maternal services, sexually transmitted infection testing and treatment services, gynaecological and urological consultations). Details of the survey methods and procedures have been described elsewhere and the life-grid method is presented in Appendix C [33,34].

### 2.3. Outcomes

Different aspects of access to SRH services (use, coverage, satisfaction and reported barriers) were considered in this analysis and were explored through the analysis of the following four outcomes: (1) use of maternal care, (2) satisfaction and difficulties to access SRH services, (3) use of modern methods of family planning (oral or injectable contraceptives, implants, male and female condoms, male or female sterilization), (4) use of HIV testing services. Use of maternal care included the use of antenatal visit and/or maternal care for giving birth and/or post-delivery care and was assessed among women who ever get pregnant. Satisfaction with SRH services was measured by asking participants to report if they experienced any difficulty the last time they used or wanted to use a SRH service and to rate with a visual scale their satisfaction with this service. The nature of the difficulties was assessed using open questions in order to not influence the participant response but responses were collected using a list of pre-determined items for better standardization. Use of family planning and HIV testing was assessed among sexual experienced participants.

### 2.4. Statistical Analysis

Conditional logistic regression was used to compare binary outcomes between people with disabilities and those without, overall and by subgroups defined by sex, the nature of the main activity limitation(s) (physical/visual/hearing) and its severity (mild versus severe or total). Mediation analysis was conducted to determine how much of the total association between the different outcomes and disability was mediated by unfavourable socioeconomic condition. The following potential mediators reflecting different aspects of the “multidimensional” poverty were considered; education level, lifetime work participation, household material wellbeing and availability of support from the social network [35]. Lifetime work participation was the proportion of the lifetime since the age of 10 years during which the participant was working or studying. A broad definition of work was used that included domestic and informal work. Household material wellbeing was measured by an index computed from household assets using the principal component analysis [36]. Availability of social support from the personal social network was measured by the number of people close to the participant who could provide support.

The decomposition of the association between disability and SRH services access outcomes into indirect (i.e., mediated by unfavourable socioeconomic condition) and direct (not mediated) effects was conducted using structural modelling to account for the sequential ordering of the mediators (Figure 1). First, the total association between disability and each SRH service access outcome was estimated using a logistic regression adjusted for age, sex and childhood socioeconomic condition. Reported experience of insufficient food and poor housing at the age of 10 years old and of not having been raised by both parents were used as surrogate measures for unfavourable socioeconomic condition during childhood. Then, the association between each SRH services access outcome and the mediating factors was assessed using logistic regression adjusted for the same factors (i.e., age, sex and childhood socioeconomic condition). Of the potential mediators, only those significantly associated with disability and with the outcome were included in the subsequent mediation analysis. In the third step, the association between the mediating factors and disability status was modelled and, then, used to estimate the direct effect of disability on the SRH service access outcome using logistic regression with inverse probability weighting (IPW) [37,38,39]. With this method, observations are weighted by the inverse of the predicted probability of disability conditional on mediating factors and adjustment factors, so that disability and mediating factors become independent. The indirect effect of the mediators was assessed in a sequentially ordered manner to account for the life-course structure of the data and post-treatment confounding assumptions [38]. Because the education level could affect the other mediating factors (work, material wellbeing and social support), it was considered as the distal factors on the pathway between disability and outcomes (Figure 1) [13,40]. Social support and material wellbeing were considered as proximal mediating factors and lifetime work participation was in between. The sequential approach allows a decomposition of the total indirect effect into the indirect effect mediated through the most distal mediating factors (e.g., education level), the remaining indirect effect mediated the next proximal mediating factor (e.g., lifetime work participation) and so forth. Confidence intervals were computed using the bootstrap method with 1000 samples. Because of the limited sample size, it was only possible to perform the mediation analysis with the overall study population and not by subgroup. Data analysis was performed using R and statistical was set to 5% [41].

All subjects gave their informed consent for inclusion before they participated in the study. The study was conducted in accordance with the Declaration of Helsinki and the protocol was approved by the “Comité d’Ethique pour la Recherche en Santé Humaine” in Cameroon (project identification code: 2014/03/431/L/CNERSH/SP) and “Comité Consultatif de Déontologie et d’Ethique” from the Institut de Recherche pour le Développement.

## 3. Results

A total of 310 persons with disabilities since childhood and 310 persons without disability of similar sex, age and residence area were included in the analysis. Characteristics of the two groups are displayed in Table 1. Compared to participants without disability, those with disabilities were living in more deprived households (*p* = 0.001), achieved lower education level (*p* < 0.001) and spent less time studying or working (*p* < 0.001). They were also more likely to have experienced insufficient food and poor housing at the age of 10 years (*p* = 0.007) and to have not been raised by their two parents (*p* = 0.005). Lastly, they reported smaller support network (*p* < 0.001). There was no evidence for a modification of the association between disability and socioeconomic factors by sex (Table 1, right column). However, as displayed in Figure 2, substantial heterogeneity was found regarding the relation between disability and education level, material wellbeing, lifetime work participation and availability of social support across the different disability sub-groups.

### 3.1. Use and Access to SRH Services

Women with disabilities since childhood were more likely than those without to have never used any SRH service (20% vs. 11%, *p* = 0.04). However, when the analysis was restricted to women who ever get pregnant, no significant difference between women with and without disabilities was observed (*p* = 0.9). One third of the male participants never used any SRH service and there was no significant difference between those with and without disabilities (*p* = 0.2).

The last SRH service visited by female participants was a HIV testing unit for 32%, a gynaecology unit for 26% and a maternity unit for 29% of them. For male participants, it was an HIV testing unit for 53%, a urology unit for 14% and a maternity unit for 14%. The frequencies were not significantly different between people with and without disabilities (*p* = 0.5 and 0.4 for male and female respondents respectively).

Participants with disabilities since childhood were more likely to report having experienced difficulties the last time they used a SRH service (odds ratio [OR] 1.81, 95% CI 1.06–3.08) and there was no significant difference between men and women (*p* = 0.5). Likewise, people with disabilities reported lower satisfaction scores (average difference in satisfaction scores between participants with and without disabilities (−0.70, 95% CI −1.10 to −0.30). Among participants with disabilities, those with mobility or hearing difficulties and those with severe limitations but not those with visual difficulties were at higher risk of having experienced difficulties with SRH services (Figure 3). The most common difficulty reported was related to health workers attitude (28.1% of participants with disabilities and 22.3% of those without, *p* = 0.2). Difficulties associated with physical accessibility were reported by 17.4% of participants with disabilities and 2.8% of those without (*p* < 0.001), excessive waiting time by, respectively, 19% and 15% (*p* = 0.3), difficulties associated to communication with health workers by 12.1% and 7.6% (*p* = 0.1) and difficulties associated to cost by 15.4% and 12%, respectively (*p* = 0.4). Participants with mobility difficulties were more likely to report difficulties associated with physical accessibility (OR 9.47, 95% CI 4.66–19.25) and those with hearing difficulties to report communication problems (OR 5.62, 95% CI 2.47–12.78).

Education level, lifetime work participation and material wellbeing were not associated with the experience of difficulties to use SRH services or with the satisfaction score (Table 2) and, therefore, were not further considered in the mediation analysis. Smaller social network that could provide support was associated with both the satisfaction score and the experience of difficulties. 

However, as displayed in Figure 4, there was no evidence that the association between disability and difficulties to use SRH services could be mediated through social network (*p* = 0.9).

### 3.2. Use of Voluntary HIV Testing

People with disabilities since childhood were significantly less likely than people without disabilities to have ever been tested for HIV (OR 0.60, 95% CI 0.39–0.92). Although fewer men than women get tested for HIV (61% vs. 82%, *p* < 0.0001), there was no evidence for a modification of the association between disability and the likelihood of HIV test by sex (*p* = 0.9). The highest risk of never having been tested was observed among participants with mobility limitation and among those with more severe limitations (Figure 3). Of the potential mediators, only education and lifetime work participation were significantly associated with both disability and the likelihood of HIV test (Table 2) and therefore included in the subsequent mediation analysis. After adjusting for education level and lifetime work participation, the association between disability and the likelihood of ever having been tested reduced by 1.5 and was no more significant, thereby indicating that a large part of the total association was mediated by these factors (Figure 4). The odds ratio for an indirect association mediated through education level was 0.82, 95% CI 0.71–0.94. In addition, there was some evidence for an indirect effect mediated through the lifetime work participation independently of the education level (OR 0.86, 95% CI 0.69–1.04, Figure 4).

### 3.3. Use of Modern Family Planning

People with disabilities since childhood were less likely than those without to ever have used any modern method of family planning (respectively, 71% vs. 79%, OR 0.56, 95% CI 0.36–0.88). There was no evidence for an interaction between disability and sex (*p* = 0.7). In sub-group analyses, less frequent use of modern family planning methods was reported by participants with mobility difficulties and by those with more severe limitations but not by those with visual or hearing difficulties (Figure 3). The use of modern family planning was also associated to participants’ education level and lifetime work participation but not to the economic score or the social support available (Table 2). An important part of the association between disability and the use of family planning methods was mediated through education level and lifetime work participation as showed in Figure 3: the odds ratio for a direct association between disability and family planning use reduced to 0.91, 95% CI 0.85–1.01 while the natural indirect effect mediated through education level on the odds ratio scale was 0.82, 95% CI 0.74–0.91. The odds ratio for the indirect effect mediated through lifetime work participation independently to the actual education level achieved was 0.86, 95% CI 0.74–0.95.

## 4. Discussion

This study examines the intersection of disability and access to SRH services in Cameroonian adults living in an urban environment. While our results confirm that men and women with disabilities since childhood are at increased risk of poorer access to SRH services, they also highlight the important variations in the disparities observed according to gender and impairment characteristics. To better understand this intersection, we investigated possible socioeconomic pathways that may explain the poorer access and use of SRH services among people with disabilities. Our finding provides good evidence that (1) men and women with disabilities since childhood have poorer education achievement and restricted access to education and work and (2) these factors explain a large part of the lower use of family planning and HIV testing in people with disabilities but not the difficulties they experienced with SRH services.

Our results bridges two lines of work showing that, on the one hand, women with disabilities have similar use of maternal care than those without disability, which has been reported in other studies [21,22,42] but that they experience more frequent difficulties and lower level of satisfaction with care [23,24,25]. This highlights that access to healthcare is a complex concept that includes multiple aspects [43]. The economic background of this study needs also to be taken into account to interpret our results. In many resource-limited settings like Cameroon where health services are not free or covered by insurance system, access to care is restricted, which might conceal some disparities [44]. This argument was used to explain the heterogeneous association between poverty and disabilities across countries of different economic levels [45,46].

This study confirms that people with disabilities—men and women—are at high risk of multidimensional poverty, which includes lack of education, employment, material resource and social support [35,45]. This comprehensive definition of poverty that goes beyond the single lack of economic resources was important to better understand the situation of people with disabilities. For instance, while the household welfare score was significantly lower in the disabled group, it was not associated with any of the outcomes, suggesting that social and cultural capital may be more important than only economic resources to explain access to care and prevention service. More research is however required to understand this finding as the economic score used in the analysis represents the overall household material resources and not the actual economic resources made available to people with disabilities for access to care or services in the area of SRH. Besides, this study highlights the important heterogeneity of their situation that varies significantly across sex and impairment types. For instance, people with hearing impairments achieved lower education level and social support but did not live in more deprived households, which could be explained by the limited access to sign language in Cameroon. By contrast, people with physical limitation were at higher risk of material poverty. Although simple message emphasizing the vulnerability of people with disabilities are required for advocacy directed toward decision-makers, their different needs resulting from the important heterogeneity of their situation should be considered at the stage of developing intervention for them. 

We found evidence that these socioeconomic differences account for a significant part of the differences observed in the use of two SRH services between people with and without disabilities. These results are important because they suggest future directions for interventions acting on “socioeconomic factors” specific to each type of impairment to improve the SRH of people with disabilities. Future research will be required as well to assess the extent to which intervention addressing early discrimination faced by people with disabilities to access education and/or work impacts on their SRH later.

This study borrows strength from its population-based design, the presence of a control group and the inclusion of different types of disabilities. However, potential limitation should be considered to interpret our results. First, data on childhood socioeconomic conditions, a potential confounding factor of the mediating variables, were limited. In addition, it should be noted that there could be residual confounding factors not accounted by the data collected. Lastly, there was limited power for sub-group analyses. Consequently, it was not possible to separate gender and types of activity limitation in the mediation analysis and non-significant tests of interaction should be interpreted with caution. It is also important to note that this study focused on people for which disability occurred before 10 years. Likely, their situation differs from that of people who became disabled later during their life.

## 5. Conclusions

In conclusion, this study showed that Cameroonian with disabilities since childhood have restricted access to SRH services. In addition, our results suggest that part of the restrictions but not all, results from socioeconomic factors that occur early during lifetime and that the nature of these factors varies with the type of SRH service considered.

## Figures and Tables

**Figure 1 ijerph-16-00417-f001:**
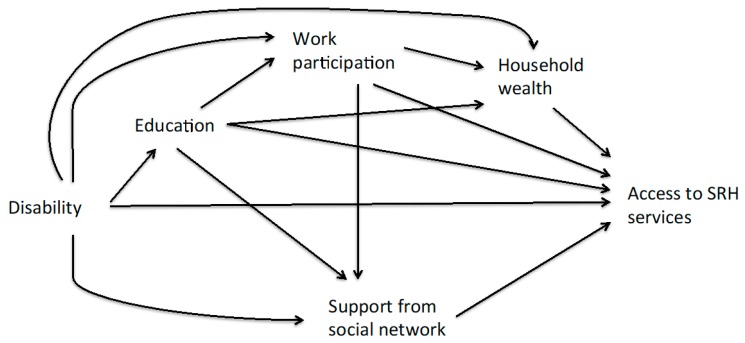
Conceptual model for direct and indirect pathways between disability and access to sexual and reproductive health (SRH) services.

**Figure 2 ijerph-16-00417-f002:**
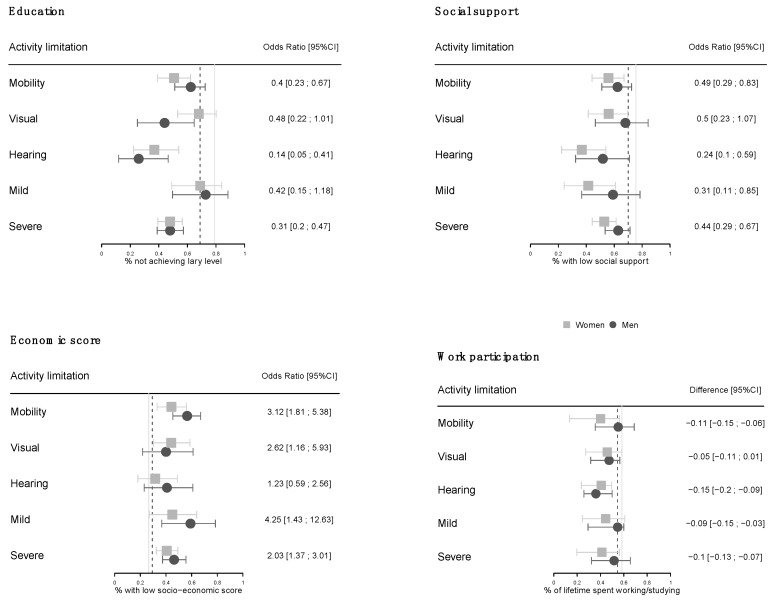
Association between disability and education level, economic status, availability of social support and participation to work by sex and type of activity limitation. Odds ratios and differences compare participants with and without disabilities of same sex, age and residential area. Vertical lines represent proportion among participants without disability (plain line: women, dotted line: men).

**Figure 3 ijerph-16-00417-f003:**
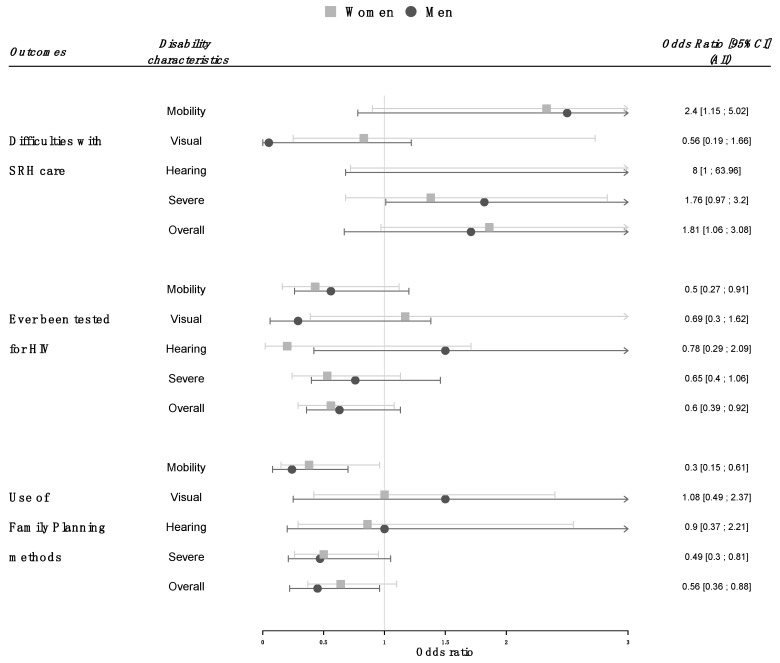
Frequencies among people with disabilities since childhood and odds ratio compared to people without disabilities of access to sexual and reproductive health services outcomes (overall, by sex and by type of activity limitations).

**Figure 4 ijerph-16-00417-f004:**
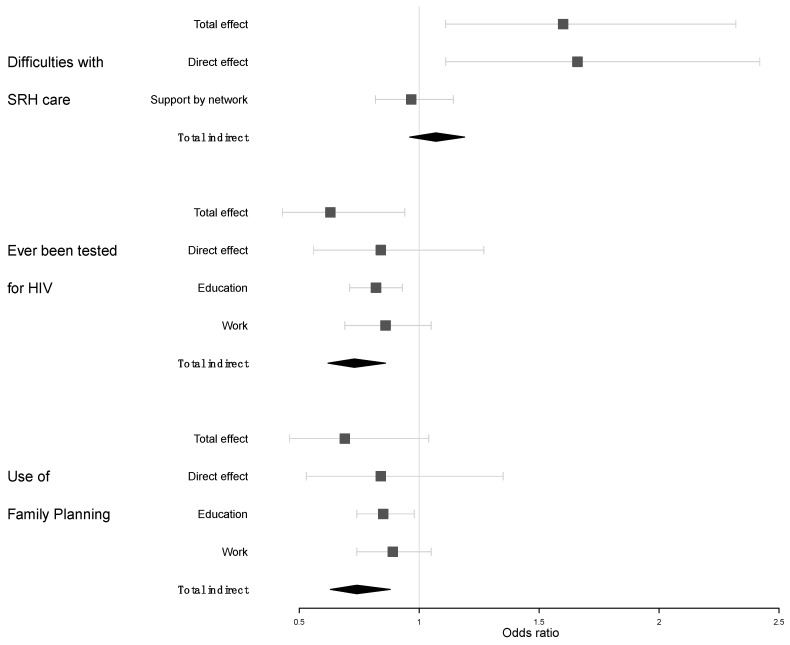
Total, direct and mediated association between disability and access to sexual and reproductive health services outcomes (vertical line indicates null effect).

**Table 1 ijerph-16-00417-t001:** Study population characteristics.

Characteristics	People with Disabilities		People with Disabilities		*p*-Value Disability by Sex Interaction ^#^
	Female (*n* = 167)	Male (*n* = 143)	Female (*n* = 167)	Male (*n* = 143)	
Age, median (IQR)	29 (21–35)	30 (23–37)	27 (22–35)	28 (23–37)	0.5
Material wellbeing score, median (IQR)	−0.52 (−1.56 to 0.60)	−0.65 (−1.50 to 0.78)	0.24 (−0.78 to 1.32)	−0.10 (−0.78 to 1.19)	0.9
Education level ^a^, *n* (%)					0.3
<Primary level	40 (24)	32 (22)	12 (7)	6 (4)	
Primary	41 (25)	37 (26)	40 (24)	24 (17)	
Secondary	67 (40)	46 (32)	80 (48)	68 (48)	
Higher	19 (11)	28 (20)	35 (21)	45 (31)	
Work situation ^b^, *n* (%)					0.4
Paid	37 (22)	52 (37)	52 (32)	71 (50)	
Informal	37 (22)	36 (25)	32 (19)	16 (11)	
Student	35 (21)	30 (21)	48 (29)	48 (33)	
No work ^c^	44 (26)	17 (12)	10 (6)	4 (3)	
Other	14 (8)	7 (5)	23 (14)	4 (3)	
Network size, median (IQR)	3 (1–5)	4 (2–7)	3 (2–6)	4 (3–7)	0.1
Lifetime work participation, median (IQR)	41 (20–56)	52 (32–62)	55 (40–62)	58 (47–68)	0.3
Poor housing/insufficient food during childhood, *n* (%)	47 (28)	34 (24)	31 (19)	21 (15)	0.3
Not raised by two parents, *n* (%)	13 (8)	10 (7)	3 (2)	5 (3)	0.9

^#^*p*-value of a test for modification of the association between the variables listed on the first column and disability status by gender; ^a^. Highest education level achieved. ^b^. Current working situation at the time of survey. ^c^. No work does not include “home work” which was included in the “other” response option. IQR: interquartile range.

**Table 2 ijerph-16-00417-t002:** Association between socioeconomic mediators and outcomes related to access to sexual and reproductive health services in male and female participants with and without disabilities.

Mediating Factors Assessed	Difficulties to Use SRH Service		Use of Family Planning Methods		Ever Been Tested for HIV	
	Male	Female	Male	Female	Female	Male
Availability of support						
No of friends/relatives who could provide support	1.06 (0.79–1.41)	1.11 (1.03–1.21)	1.19 (0.91–1.55)	0.97 (0.85–1.11)	1.00 (0.87–1.15)	0.90 (0.75–1.07)
Education level						
< Primary	Reference	Reference	Reference	Reference	Reference	Reference
Primary	3.31 (0.32–33.80)	0.86 (0.11–6.85)	0.97 (0.18–5.29)	4.42 (0.91–21.48)	1.70 (0.37–7.68)	3.70 (0.91–15.11)
Secondary	1.00 (0.13–7.60)	1.80 (0.31–10.47)	3.25 (0.68–15.56)	5.70 (1.11–29.14)	2.67 (0.61–11.72)	3.93 (0.95–16.31)
Higher	1.60 (0.22–11.91)	2.51 (0.34–18.47)	5.66 (0.83–38.63)	12.62 (1.79–88.81)	11.87 (1.63–86.40)	2.41 (0.35–16.70)
Household material wellbeing						
Economic score	0.66 (0.38–1.13)	1.03 (0.90–1.17)	1.17 (0.87–1.57)	1.07 (086–1.33)	1.44 (1.03–2.01)	1.0 (0.79–1.28)
Lifetime work participation						
Proportion of the lifetime working/studying	1.0 (0.48–2.08)	1.07 (0.63–1.82)	1.84 (1.02–3.34)	2.92 (1.49–5.70)	1.44 (0.91–2.29)	3.46 (1.44–8.35)

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
