# Peer review of "Disability and Access to Sexual and Reproductive Health Services in Cameroon: A Mediation Analysis of the Role of Socioeconomic Factors"

_ijerph, 2019, doi:10.3390/ijerph16030417_

Reviewer 1 Report

This is a very well written and conceived research study.  It is quite rare to be able to conduct a study where people with disabilities, are compared to a similar group of people without known disability.  The comparison is a powerful tool, and added strength to this paper.

That said, there are some areas that further clarification and explanation would be warranted.

1.The study group of PWD were only those who had a disability from birth (or at least within the first 10 years).  While I understand the need to make methodological decisions, it would be useful for the authors to describe this rationale, and then provide further discussion on this limitation.  The way it is written, the inclusion is described early, and then in the conclusions, with limited references throughout.  It would seem that this cohort of PWD might be different (in terms of access to SRH) that those who acquired a disability later in life.  Moreover, might the results be related to people living with outcomes (ie mobility) from developmental disabilities only?  Further description of the group would be useful.

2. While I am not that familiar with the methodology applied in this study, it seemed to me that the results indicate items that are fairly well evidenced and accepted (eg. PWDs have lower education, are well well-off economically, have worse access to education and employment), and that these factors can act as a barrier to accessing health services, including SRH.  This limited access is likely to be worse in low resource settings, where few welfare state policies apply.  The powerful potential result, as the authors indicate in the discussion, is that PWD are less satisfied when they do access care.  This hints at the perceptions and attitudes of health workers, which could be influencing the outcomes. A suggestion is to provide greater simplicity (and greater directness) in the description and discussion of the results.

3. The last line of the abstracts reads: “In conclusion, Cameroonians with disabilities since childhood have restricted access to SRH services resulting from socioeconomic factors occurring early during the life-course.” After reading through the paper a few times, it seems more to me that it is the ‘type’ socioeconomic factors in early life that have influence, not that simply that they existed.  For instance, the authors do remark that social and cultural capital might have a protective effect on these outcomes.  Can the authors further clarify?

4. To really understand the meaning of these results (e.g., in the discussion), maybe a brief description of how and where a PWD would access care in Yolande.

5. How was Figure 1 developed, based on which theory (or maybe no theory, which is okay).  Just a little more for context.

5. Minor: Table 1 seems to be missing the ‘with’ vs. ‘without’ disabilities description in the heading.

Author Response

Reviewer 1.

This is a very well written and conceived research study.  It is quite rare to be able to conduct a study where people with disabilities, are compared to a similar group of people without known disability.  The comparison is a powerful tool, and added strength to this paper.

Thank you

That said, there are some areas that further clarification and explanation would be warranted.

1.The study group of PWD were only those who had a disability from birth (or at least within the first 10 years).  While I understand the need to make methodological decisions, it would be useful for the authors to describe this rationale, and then provide further discussion on this limitation.  The way it is written, the inclusion is described early, and then in the conclusions, with limited references throughout.  It would seem that this cohort of PWD might be different (in terms of access to SRH) that those who acquired a disability later in life.  Moreover, might the results be related to people living with outcomes (ie mobility) from developmental disabilities only?  Further description of the group would be useful.

Response: The choice to restrict the analysis to people with disabilities onset before the age of 10 years came from two lines of arguments. We agree with the reviewer that the situation of the people who grew up with disabilities most likely differs from that of people who acquired a disability later. Mixing the two populations would have blurred the results, which led us to focus on the first group. Moreover, the cross-sectional design of this study using only retrospective information on the occurrence of different events precludes a rigorous mediation analysis of the direct and indirect effects of disability on access to SRH services because the chronological ordering of exposure and mediator is not guaranteed.

In this revised version, more reference to the inclusion criteria referring to the age of disability has been included. In addition, we have included in the discussion a mention to the focus of this analysis on people with disabilities before 10 years. Now read: “It is also important to note that this study focused on people for which disability occurred before 10 years. Likely, their situation differs from that of people who became disabled later during their life”.

In this study, the identification of the participants with disabilities was based on the presence of activity limitation (identified using the Washington Group disability questionnaire). Therefore, as participants were older than 15 years and interviewers did not have the competence to make retrospectively a diagnosis on the cause of disability, we are to provide more detailed description of the cause of disability. 

2. While I am not that familiar with the methodology applied in this study, it seemed to me that the results indicate items that are fairly well evidenced and accepted (eg. PWDs have lower education, are well well-off economically, have worse access to education and employment), and that these factors can act as a barrier to accessing health services, including SRH.  This limited access is likely to be worse in low resource settings, where few welfare state policies apply.  The powerful potential result, as the authors indicate in the discussion, is that PWD are less satisfied when they do access care.  This hints at the perceptions and attitudes of health workers, which could be influencing the outcomes. A suggestion is to provide greater simplicity (and greater directness) in the description and discussion of the results.

Response: We are sorry that the reviewer felt the presentation of our results was too complicated. Our primary aim in this work was to conduct a mediation analysis to better understand the pathways involved in the production of inequities in people with disabilities in the specific context of Cameroon. Although mediation analysis is a very powerful tool, there are a number of pitfalls to avoid in such analysis (e.g. see Ten Have, 2012 [1] or Richiardi, 2013 [2]). This may be a reason for the complexity noted by the reviewer. The two sections have been reviewed careful by the authors to improve their readability.

3. The last line of the abstracts reads: “In conclusion, Cameroonians with disabilities since childhood have restricted access to SRH services resulting from socioeconomic factors occurring early during the life-course.” After reading through the paper a few times, it seems more to me that it is the ‘type’ socioeconomic factors in early life that have influence, not that simply that they existed.  For instance, the authors do remark that social and cultural capital might have a protective effect on these outcomes.  Can the authors further clarify?

Response: The reviewer is correct that the nature of the socioeconomic factors found associated with restricted access to SRH service depends of the nature of the SRH service considered. We have amended the conclusion to clarify this point. Now read: “In addition, our results suggest that these restrictions result from socioeconomic factors that occur early during lifetime and that the nature of these factors varies with the type of SRH service considered.”

4. To really understand the meaning of these results (e.g., in the discussion), maybe a brief description of how and where a PWD would access care in Yolande.

Response: A large range of SRH services were considered in this work and the study included participants from all over Yaoundé, capital of Cameroon. Therefore, it would be extremely difficult to give a correct description of how and where people could access care in Yaoundé. In the discussion we mentioned that “health services are not free or covered by insurance system”, in Cameroon (line 271). In addition we have included a reference to the excellent book edited by Kuate-Defo that gives a comprehensive overview of the sexual and reproductive health issues and care in Cameroon.

5. How was Figure 1 developed, based on which theory (or maybe no theory, which is okay).  Just a little more for context.

Response: Figure 1 is based on the work by Singh-Manoux and colleagues and on the work of the Commission on the Social Determinant of Health (CSDH) who consider a causal and temporal ordering of the social determinants. Accordingly, social determinants are structured into proximal and distal factors, the former structuring the later. Reference to this conceptual framework is now included in the revised version of the manuscript.

6. Minor: Table 1 seems to be missing the ‘with’ vs. ‘without’ disabilities description in the heading.

The typo has been corrected.

References

1.         Ten Have, T.R.; Joffe, M.M. A review of causal estimation of effects in mediation analyses. Statistical Methods in Medical Research 2012, 21, 77-107

            2.         Richiardi, L.; Bellocco, R.; Zugna, D. Mediation analysis in epidemiology:             methods, interpretation and bias. International journal of epidemiology 2013, 42,             1511-1519  

Reviewer 2 Report

This study aimed to investigate the relationships between disability and access to and use of SRH services among people with disabilities in Cameroon. 
The manuscript is clear and detailed although the statistical analyses may hard. 

I had only one concern in the information were lacked about the study approvement of institutional review board. Maybe it had report in other article at the same study? But I thank it should also report again in this study.

Author Response

This information is provided page 4 lines 155 to 159: “All subjects gave their informed consent for inclusion before they participated in the study. The study was conducted in accordance with the Declaration of Helsinki, and the protocol was approved by the “Comité d’Ethique pour la Recherche en Santé Humaine” in Cameroon, and “Comité Consultatif de Déontologie et d’Ethique” from the Institut de Recherche pour le Développement (IRD).”  

Reviewer 3 Report

Thank you for the opportunity to review your article

REVIEW REPORT

Brief summary

The aim of the paper is to measure the access of people with disabilities to SHR services. In the context of health inequities, this study has the merit to define the several aspects and reasons regarding the possibilities of people with disability to access to essential health services. Considering the large part of population who live with disability in the considered context, the approach of the study is particularly important for the public health point of view in order to plan strategies to facilitate the necessary access to the services of a significant part of population.

Broad comments

a) Areas of  strength

The reasons of the poorer access to the SRH services of individuals with disability are widely addressed. The analyses of the factors influencing the access are well defined and the comparison with individuals without disabilities completely verified. The results of differences, although expected, are deeply and clearly measured thanks to an interesting and organized methodology.

b) Areas of weakness

The presentation of some results explained in the article should be improved in the description both in the text and in the tables/figures. For example, not always is so clear data described in the text and the table of figure where they are presented. A point by point analysis is explained in the specific comments. Sometimes the presentation of The data reported in the text is not always clear compared to those shown in the tables or figures. A better and simpler description is necessary to better represent the results of the study and the relation between disability and the impact of the different variables in the different use of RSH. Another point that the authors could improve is the citations of studies of literature that have same results in the consideration of the relationship between some variables or health determinants and the access to health services with a specific regard to the importance of education level as a proxy of other social health determinants (social and economic conditions).

Specific comments

As for the single aspects to be changed or improved, please see below our comments to the single lines of article.

35: add in the world to “An estimated of one billion people” 50-63: verify if put these part in discussion as comparison with your results

62: insert a space between “barriers” and the citation.

80: how the authors conducted the multistage sampling strategy to select random the people for the study?

81: eliminate a space before “All people”

83: the authors should describe better and with more details the questionnaire used to define the impairment of the population selected for the study and put citations about its validation

96: Describe briefly the method

97: Eliminate a space before (maternal service…

146: The authors could cite some studies about the central role of education as a proxy of other social health determinants (work condition, social status, income) and refer their affirmations to the literature

154: What is the level of statistical significance established for the study? Declare it

171: Maybe the last right column?

175: Table 1: in the first column on the left, what is the with the two “<Primary level” under” Work situation”

Add a space between n and ( at the line of work situation

178: Where is this abbreviation SHR  in the table? If it isn't necessary eliminate it

179: Figure 2:  Use a bigger character for all the written parts of the figure and put the legend in a greater character and in a clearer position

179: Figure 2: In the first graphic on the left “Education”, What does “lary level” mean?

183: among for amon

185: Access without to

187: Which statistical test did you use to compare the data?

200: Put ( before -0.70

205: It is sufficient write p<0.001< span="">

210: Data don't correspond with those indicated in figure 3

Figure 3: Use a greater character in all part of the table

216: These association aren't so clear seeing the table 2 where data are stratified for sex and the OR risulted over the 1.0 with differences between males and females. Please, can the authors explain better their affirmations?

 218:  Put table 2 before the phrase: “However, as displayed in Figure 4, there was no evidence that the association………”

Table 2: Do data regard the participants  with disabilities? Insert in the title of the table

What is Ref?

Put 2.67 in the column female of Ever been tested for HIV, at line secondary of education level, over the data in brackets, like the other 

221: Figure 4: Use a bigger character in all part of the table

240: Add (Figure 4) after data of education level.

272: The authors should cite studies about the central role of education level as proxy of the impact of other social determinants (type of work, income, housing) on the health inequities that this study confirm.

298: Have you information about the occupational condition of the subjects included in the study, with or without disability? Describe better, also in other part of the article, what are the different level of work situation of the table 1. A key to address the difficulties of people with disabilities is in the multidimensional method of Disability Management that in other countries is growing up in order to improve the use of services and the work activity for people with disability and/or limitations. As for this argument, I suggest to insert the citations of these two studies:

a)      Camisa V, Vinci MR, Santoro A, Brugaletta R, Zaffina S, Apostoli P. Disability Management: National and international context. G Ital Med Lav Erg 2016;38:3, 224-227

b)      Camisa V, Vinci MR, Santoro A, Dalmasso G, Bianchi N, Raponi M, Brugaletta R, Derrico P, Zaffina S. Disability Management in a complex health care facility: management activities of the competent doctor. G Ital Med Lav Erg 2016;38:3, 228-231

308: Add a “,” after disabilities and childhood

Author Response

Reviewer 3.

Brief summary

The aim of the paper is to measure the access of people with disabilities to SHR services. In the context of health inequities, this study has the merit to define the several aspects and reasons regarding the possibilities of people with disability to access to essential health services. Considering the large part of population who live with disability in the considered context, the approach of the study is particularly important for the public health point of view in order to plan strategies to facilitate the necessary access to the services of a significant part of population.

Broad comments

a) Areas of  strength

The reasons of the poorer access to the SRH services of individuals with disability are widely addressed. The analyses of the factors influencing the access are well defined and the comparison with individuals without disabilities completely verified. The results of differences, although expected, are deeply and clearly measured thanks to an interesting and organized methodology.

b) Areas of weakness

The presentation of some results explained in the article should be improved in the description both in the text and in the tables/figures. For example, not always is so clear data described in the text and the table of figure where they are presented. A point by point analysis is explained in the specific comments. Sometimes the presentation of The data reported in the text is not always clear compared to those shown in the tables or figures. A better and simpler description is necessary to better represent the results of the study and the relation between disability and the impact of the different variables in the different use of RSH. Another point that the authors could improve is the citations of studies of literature that have same results in the consideration of the relationship between some variables or health determinants and the access to health services with a specific regard to the importance of education level as a proxy of other social health determinants (social and economic conditions).

Response: the manuscript has been reviewed carefully, all comments below have been addressed and simplification made when possible.

Specific comments

As for the single aspects to be changed or improved, please see below our comments to the single lines of article.

35: add in the world to “An estimated of one billion people”

Response: The change has been done.

50-63: verify if put these part in discussion as comparison with your results

Response: The authors prefer to keep these sentences in the introduction as they provide the rational for conducting this analysis.

62: insert a space between “barriers” and the citation.

Response: The typo has been corrected.

80: how the authors conducted the multistage sampling strategy to select random the people for the study?

Response: The multistage sampling strategy has been described in a comprehensive way in a former paper (Debeaudrap et al. Open BM…) and provided in this paper in appendix 1. This description was not initially included as we believed it was out of the scope of this paper. Should our paper be accepted, we leave the inclusion of this appendix as supplementary material to the Editors discretion.

81: eliminate a space before “All people”

Response: The change has been done.

83: the authors should describe better and with more details the questionnaire used to define the impairment of the population selected for the study and put citations about its validation

Response: The description of the Washington group questionnaire along with reference has been included in appendix 2.

96: Describe briefly the method

Response: The description of the life-grid method has been included in appendix 3.

97: Eliminate a space before (maternal service…

Response: The change has been done.

146: The authors could cite some studies about the central role of education as a proxy of other social health determinants (work condition, social status, income) and refer their affirmations to the literature

Response: we have included references on the proximal role of education among the social determinants of health (Singh-Manoux A, 2002 [1] and CSDH, 2008 [2]). However, it is important to highlight that, in this paper, we considered education to be a proximal factor and not a proxy for other social determinants. In other words, our conceptual approach was to assume causal ordering and we postulated that education occurs before the other factors (e.g. work) and affects them.

154: What is the level of statistical significance established for the study? Declare it

Response: The statistical significance was 5%. This has been included in the Methods section (line 183).

171: Maybe the last right column?

Response: ‘left” has been changed to “right” in the text

175: Table 1: in the first column on the left, what is the with the two “<Primary level” under” Work situation”

Response: The labels have been corrected.

Add a space between n and ( at the line of work situation

Response; described change has been made.

178: Where is this abbreviation SHR  in the table? If it isn't necessary eliminate it

Response: the sentence has been removed from the footnote.

179: Figure 2:  Use a bigger character for all the written parts of the figure and put the legend in a greater character and in a clearer position

The Figure has been updated accordingly.

179: Figure 2: In the first graphic on the left “Education”, What does “lary level” mean?

Response: Iary stands for primary. It has been changed in the text.

183: among for amon

The typo has been corrected.

185: Access without to

The typo has been corrected

187: Which statistical test did you use to compare the data?

Response: conditional logistic regression as described line 134 in the method section.

200: Put ( before -0.70

Response: bracket has been added

205: It is sufficient write p<0.001< span="">

Response; change made.

210: Data don't correspond with those indicated in figure 3

Response: Indeed there was an ambiguity with the term accessibility. Figure 3 display results for general accessibility to SRH services while results line 210 is for physical accessibility. The text has been changed in the revised version to remove this ambiguity. 

Figure 3: Use a greater character in all part of the table

The font size has been increased in the new Figure.

216: These association aren't so clear seeing the table 2 where data are stratified for sex and the OR risulted over the 1.0 with differences between males and females. Please, can the authors explain better their affirmations?

Response: All the 95%CI included 1, which means that the ORs were not statistically different from 1.

 218:  Put table 2 before the phrase: “However, as displayed in Figure 4, there was no evidence that the association………”

Response: The change has been made.

Table 2: Do data regard the participants with disabilities? Insert in the title of the table

Response: This analysis was conducted on the full dataset (i.e. participants with and without disabilities). This information has been included in the caption.

What is Ref?

Response: “Ref” means that this is the reference category. The term has been change to “reference” in the revised manuscript.

Put 2.67 in the column female of Ever been tested for HIV, at line secondary of education level, over the data in brackets, like the other 

Response: the correction has been made as suggested.

221: Figure 4: Use a bigger character in all part of the table

Response: greater font size has been used in the revised Figure.

240: Add (Figure 4) after data of education level.

Response: The suggested change has been made.

272: The authors should cite studies about the central role of education level as proxy of the impact of other social determinants (type of work, income, housing) on the health inequities that this study confirms.

Response: we have included two references (see above). However, it should be noted that our results do not provide evidence on the fact that education is a proxy for other social determinants. Instead we assumed causal ordering and postulated (from other references such as those cited above) that education acts as a proximal factor. In other words, education is considered as a factor that occurs early and affects the other factors (e.g. work).

298: Have you information about the occupational condition of the subjects included in the study, with or without disability? Describe better, also in other part of the article, what are the different level of work situation of the table 1. A key to address the difficulties of people with disabilities is in the multidimensional method of Disability Management that in other countries is growing up in order to improve the use of services and the work activity for people with disability and/or limitations. As for this argument, I suggest to insert the citations of these two studies:

Response: Thank you for the suggestion and the references. Unfortunately, we are only able to read the abstracts as the papers are written in Italian.

Information on participants occupations is not detailed because our focus was to contrast the situation of 1) people working and paid, 2) working in the informal sector, 3) not working (the student being apart).

308: Add a “,” after disabilities and childhood

Response: The change has been made.

References

1.         Singh-Manoux, A.; Clarke, P.; Marmot, M. Multiple measures of socio-economic position and psychosocial health: proximal and distal measures. International journal of epidemiology 2002, 31, 1192-1199; discussion 1199-1200.

2.         WHO. Closing the gap in a generation: health equity through action on the social determinants of health. Final report of the Commission on the Social Determinants of Health. Availabe online: http://www.who.int/social_determinants/final_report/csdh_finalreport_2008.pdf (accessed on September 2016).